# Haplopine Ameliorates 2,4-Dinitrochlorobenzene-Induced Atopic Dermatitis-Like Skin Lesions in Mice and TNF-α/IFN-γ-Induced Inflammation in Human Keratinocyte

**DOI:** 10.3390/antiox10050806

**Published:** 2021-05-19

**Authors:** Tae-Young Kim, Ye Jin Kim, Jonghwan Jegal, Beom-Geun Jo, Han-Seok Choi, Min Hye Yang

**Affiliations:** 1College of Pharmacy, Pusan National University, Busan 46241, Korea; taeyour@pusan.ac.kr (T.-Y.K.); jhjegal@pusan.ac.kr (J.J.); bg_jo@pusan.ac.kr (B.-G.J.); 2C&D Research Team, R&D Strategy Center, Genuonesciences, Seoul 06800, Korea; yejin.kim@genuonesciences.com

**Keywords:** haplopine, atopic dermatitis, HaCaT cells, Jurkat T cells, 2,4-dinitrochlorobenzene, Balb/c mice

## Abstract

This study aimed to investigate the anti-inflammatory, antioxidant, and anti-atopic dermatitis (AD) effects of haplopine, which is one of the active components in *D. dasycarpus.* Haplopine (12.5 and 25 μM) inhibited the mRNA expressions of inflammatory cytokines IL-6, TSLP, GM-CSF, and G-CSF and the protein expressions of IL-6 and GM-CSF in TNF-α/INF-γ-stimulated HaCaT cells. In H_2_O_2_-induced Jukat T cells, haplopine (25 and 50 μM) suppressed the productions of proinflammatory cytokines (IL-4, IL-13, and COX-2) and increased the mRNA and protein expressions of oxidative stress defense enzymes (SOD, CAT, and HO-1) in a concentration-dependent manner. In vivo, haplopine significantly attenuated the development of AD symptoms in 2,4-dinitrochlorobenzene (DNCB)-stimulated Balb/c mice, as evidenced by reduced clinical dermatitis scores, skin thickness measurements, mast cell infiltration, and serum IgE concentrations. These findings demonstrate that haplopine should be considered a novel anti-atopic agent with the potential to treat AD.

## 1. Introduction

Atopic dermatitis (AD) is one of the most common inflammatory skin diseases. AD affects all ages, from newborns to adults, and its prevalence has increased sharply over the past four decades, especially in developed countries [1]. The symptoms of AD include intensive pruritus, eczematous changes, skin thickening, and skin degradation [2]. Quality of life is markedly impaired in AD patients as it disturbs sleep, lowers physical activity, and adversely affects psychological health [1,2]. AD is the result of complex interactions between genes involved in the skin barrier, immunological dysregulation, and environmental factors [3,4,5] and is classified as extrinsic or intrinsic according to the presence or absence of the immune response associated with IgE specific to an external antigen. In these different disease types, the cells and cytokines dominantly involved differ, but both result in an inflammatory response [6], which suggests anti-inflammatory agents may be effective treatments regardless of the AD type.

Reactive oxygen species (ROS) are inevitable by-products of metabolism in all organisms that require oxygen. ROS are highly reactive and, when produced in appropriate amounts, prevent microorganism invasion and contribute to immunity and intracellular signaling. However, when excessive, ROS can cause oxidative stress and destroy cellular components, such as DNA, proteins, and lipid bilayers. To prevent these effects, organisms activate an antioxidant system to remove excess ROS and achieve redox equilibrium [7]. Many studies on AD have shown that oxidative stress plays an important role in the development of AD. Skin acts as a barrier that prevents microorganism invasion, but when lipid bilayers of cell membranes are disrupted by oxidative stress, foreign agents can invade tissues and induce inflammatory responses that increase the risk of AD [8]. Previous studies revealed increased oxidative stress and altered antioxidant defense as important contributing factors in the pathogenesis of AD [9,10,11]. Furthermore, several clinical trials have shown that antioxidative nutrients, such as vitamins A, C, and E, alleviate the symptoms of AD [9,12].

*Dictamnus dasycarpus* Turcz. belongs to the family Rutaceae and is a perennial herbaceous plant widely distributed in Asia. Its root bark, called “Baekseonpi” in Korean, has been used to treat mild skin diseases in traditional medicine in Korea, China, and Japan [13]. *D. dasycarpus* extracts have been reported to have anti-atopic [14], anti-inflammatory [15], and antioxidant [15], anti-allergic [14] effects, and alkaloids (e.g., dictamine and haplopine) and limonoids (e.g., fraxinellone and obacunone) isolated from *D. dasycarpus* have been shown to possess anti-inflammatory [16,17,18], antioxidative [18,19,20], and antifungal [21] activities. Although many studies have been conducted on the bioactivity of *D. dasycarpus*, few have investigated the use of its constituent compounds for the treatment of AD. In this study, we investigated the anti-inflammatory effects of dictamine, fraxinellone, haplopine, and obacunone and the anti-AD effect of haplopine in human HaCaT and Jurkat T cells and in a male Balb/c mouse DNCB (2, 4-dinitrochlorobenzene)-induced model of AD and the mechanisms involved.

## 2. Materials and Methods

### 2.1. Materials

4,8-Dimethoxy-9H-furo[2,3-b]quinolin-7-one (haplopine) was purchased from BioBioPha (Kunming, Yunnan, China), and dictamine, obacunone, and fraxinellon were from Avention Corporation (Seoul, Korea) (Figure 1). DNCB, DMSO (dimethyl sulfoxide), tacrolimus, DPPH (2,2-diphenyl-1-picrylhydrazyl), and DCFH-DA (2′,7′-dichlorofluorescein-diacetate) were purchased from Sigma-Aldrich (St. Louis, MO, USA). Dulbecco’s modified Eagle’s Medium (DMEM), fetal bovine serum (FBS), penicillin, streptomycin, and trypsin were from Gibco-BRL (Grand Island, NY, USA). All primers were purchased from Bioneer (Daejeon, South Korea), and Trizol solution, cDNA Synthesis kits, and SYBR solution were from PhileKorea (Seoul, Korea). α-Tubulin and catalase (CAT) were from Cell Signaling Technology (Boston, MA, USA), and anti-Cox-2, anti-SOD-1, and anti-HO-1 antibodies were purchased from Santa Cruz Biotechnology Inc. (Santa Cruz, CA, USA). All other reagents used were of the purest grades available.

### 2.2. Cell Culture

HaCaT cells (a human keratinocyte cell line) and Jurkat T cells (a human T lymphocyte cell line) were cultured in high glucose Dulbecco’s modified Eagle’s medium or RPMI 1640 medium (Gibco Laboratories, Grand Island, NY, USA) containing 10% heat-inactivated FBS, 100 units/mL penicillin, and 100 μg/mL streptomycin (Invitrogen, Carlsbad, CA, USA) in a humidified 5% CO_2_ atmosphere at 37 °C. Media were changed every 2 days during incubation.

### 2.3. Quantitative Real-Time PCR

HaCaT or Jurkat T cells were seeded in 100 mm culture plates at a density of 1 X 10^6^ cells per plate. After incubation for 24 h at 37 °C, HaCaT or Jurkat T cells were pretreated with haplopine at 12.5, 25, and 50 μM for 0.5 h. TNF-α 10 ng/mL plus IFN-γ 10 ng/mL or H_2_O_2_ 50 μM were then added to the haplopine-pretreated cells, and the cells were harvested 24 h or 48 h later. The total RNA was then isolated using Trizol RNA extraction reagent, and 1 μg of total RNA was reverse transcribed using the cDNA Synthesis Kit. The cDNAs obtained were used as quantitative real-time PCR templates. The primers used for real-time PCR were designed using BLAST, as follows: mRNA encoding interleukin-6 (IL-6, Gene ID 3569) forward primer 5′-aaagaggcactgccagaaaa-3′ and reverse 5′-atctgaggtgcccatgctac-3′, interleukin-4 (IL-4, Gene ID 3565) forward primer 5′-tgcctccaagaacacaactg-3′ and reverse primer 5′-ctctggttggcttccttcac-3′, interleukin-13 (IL-13, Gene ID 3596) forward primer 5′-agccaacgagtaatttattgtttttc-3′ and reverse primer 5′-aactttatttctggcttcagtttgat -3′, COX-2 (Gene ID 4513) forward primer 5′-ccttcctcctgtgcctgatg-3′ and reverse primer 5′-acaatctcatttgaatcaggaagct-3′, SOD-1 (Gene ID 6647) forward primer 5′-gggagatggcccaactactg-3’ and reverse primer 5′-ccagttgacatgcaaccgtt-3’, CAT (Gene ID 847) forward primer 5′-atggtccatgctctcaaacc-3′ and reverse primer 5′-caggtcatccaataggaagg-3’, HO-1 (Gene ID 3162) forward primer 5′-aagactgcgttcctgctcaac-3′ and reverse primer 5′-aaagccctacagcaactgtcg-3′, Nrf2 (Gene ID 4780) forward primer 5′-tactcccaggttgcccaca-3′ and reverse primer 5’-catctacaaacgggaatgtctgc-3′, thymic stromal lymphopoietin (TSLP, Gene ID 85480) forward primer 5’-gcttcctgtggactggcaat-3′ and reverse primer 5′-aaggcaaaagggaacatacg-3′; granulocyte-macrophage colony-stimulating factor (GM-CSF, Gene ID 1437) forward primer 5′-ttctgcttgtcatccccttt-3′ and reverse primer 5′-cttctgccatgcctgtatca-3′; granulocyte colony-stimulating factor (G-CSF, Gene ID 1440) forward primer 5′-ccccatcccatgtatttatctctatt-3’, reverse primer 5′-tggtatttacctatctacctcccagt-3′; GAPDH (Gene ID 2597)forward primer 5’-agggctgcttttaactctggt-3’ and reverse primer 5’-ccccacttgattttggaggga-3’. Quantitative real-time PCR was conducted using Universal (SYBR Green) qPCR master mix (New England Biolabs, Ipswich, MA, USA), and qPCR was performed using a QuantStudio3 Real-time PCR Detection System (Applied Biosystems, Waltham, MA, USA) programmed at 40 cycles of 95 °C for 15 s, 60 °C for 60 s, and 72 °C for 40 s. Relative mRNA levels of genes were normalized versus GAPDH mRNA.

### 2.4. ELISA Assay

The HaCaT cells were cultured in 100 mm culture plates (1 × 10^6^ cells/well) for 24 h. HaCaT cells were pretreated with haplopine at 12.5 and 25 μM for 0.5 h. TNF-α 10 ng/mL plus IFN-γ 10 ng/mL were then added to the haplopine-pretreated cells, and the supernatant was harvested 24 h later. The concentration of IL-6 and GM-CSF were measured using enzyme-linked immunosorbent assays (ELISA) according to the manufacturer’s instructions. (R&D System, Wiesbaden, Germany). The absorbance was measured at 450 nm using an ELISA reader (TECAN, Infinite F200 pro, Männedorf, Switzerland).

### 2.5. Western Blot Analysis

Protein expression was assessed by Western blot analysis according to standard procedures. The Jurkat T cells were cultured in 100 mm culture plates (1 × 10^6^ cells/well). After incubation for 24 h at 37 °C, Jurkat T cells were pretreated with haplopine at 12.5, 25, and 50 μM for 0.5 h. TNF-α 10 ng/mL plus IFN-γ 10 ng/mL were then added to the haplopine-pretreated cells, and the cells were harvested 24 h later. The cells were washed twice in ice-cold PBS (pH 7.4). The cell pellets were suspended in a lysis buffer on ice for 20 min, and the cell debris was removed by centrifugation. Protein concentrations were determined using a Bradford protein assay reagent (Bio-Rad Laboratories, Hercules, CA, USA) according to the manufacturer’s instructions. Equal amounts of protein were subjected to sodium dodecyl sulfate-polyacrylamide gel electrophoresis and then transferred onto an iBolt 2 PVDF membrane regular stacks (Invitrogen, Carlsbad, CA, USA). The membrane was blocked with 5% nonfat milk in Tris-buffered saline with Tween-20 buffer (150 mM NaCl, 20 mM Tris-HCl, and 0.05% Tween-20, pH 7.4). After blocking, the membrane was incubated with primary antibodies (1:1000 dilution) at 4 °C for 24 h, washed with Tris-buffered saline with Tween-20, and incubated again with anti-mouse immunoglobulin G horseradish peroxidase-conjugated secondary antibodies (1:5000 dilution) for 2 h at room temperature. Immunoreactive bands were detected using SuperSignal WestPico chemiluminescence substrate (Thermo Fisher Scientific, Waltham, MA, USA).

### 2.6. Experimental Animals

Six-week-old male Balb/c mice (6w, male) were obtained from Samtako Bio Korea. All mice were housed in a special pathogen-free room, fed mouse chow, and provided standard water *ad libitum*. Experiments were performed in a controlled environment (23 ± 2 °C, RH 50 ± 10%, under a 12/12-h light/dark cycle and 10–18 air circulation changes/h). Animals were divided randomly into five groups of seven mice; a normal control group (Con), a DNCB-treated group (negative controls, NC), a tacrolimus-treated group (positive controls, PC), a 0.05% haplopine-treated group (Haplopine 0.05%), or a 0.1% haplopine-treated group (Haplopine 0.1%). All animal studies were performed according to the guidelines issued by the Ethics Committee for the Use of Experimental Animals at Kolmar Korea Co., Ltd. (certification number: 19-NP-AT-003-P).

### 2.7. AD Induction and Treatment

To induce AD in mice, we used DNCB (Sigma-Aldrich) sensitization followed by the DNCB challenge. Briefly, the backs of mice were shaved with an electric clipper a day before sensitization with DNCB solution (acetone:olive oil = 3:1 *v*/*v*). For sensitization, a 1 cm × 1 cm gauze-attached patch containing 200 μL of 2% (*w*/*v*) DNCB was applied to the shaved area twice weekly. One week after sensitization, dorsal skin was challenged with 200 μL of a 0.5% DNCB solution twice weekly. After the DNCB challenge, haplopine (0.05% or 0.1%) was administered daily for 2 weeks. Positive controls were treated with 50 mg of protopic (0.1% tacrolimus) daily for 2 weeks.

### 2.8. Clinical Dermatitis Score

The scoring was based on the severities of erythema/hemorrhage, edema, excoriation/erosion, and dryness/scarring/inflammation of dorsal skin, which were scored as 0 or 1. Dermatitis scores were calculated by summing scores for these four signs (none = 0; mild = 1; moderate = 2; severe = 3).

### 2.9. Measurement of Serum IgE Levels

Blood samples were collected from abdominal aortas, and the serum was separated by centrifugation at 1500 rpm for 30 min and stored at −80 °C until required. Serum IgE concentrations were measured using an IgE ELISA kit (Becton and Dickinson, Franklin Lakes, NJ, USA), according to the manufacturer’s instructions.

### 2.10. Spleen and Body Weight

Mouse body weights were measured before sacrifice, and the weights of the spleens were measured using an electronic balance.

### 2.11. Histopathological Analysis

Skin tissue slices were fixed in 10% neutral buffered formalin (BIOSESANG, Gyeonggi-do, Korea) for 24 h at 4 °C, paraffin-embedded, sectioned, and stained with hematoxylin and eosin (H&E) or toluidine blue (TB) for the detection of the epidermal thickness and mast cells, respectively. Images were captured using an Olympus DP controller and manager at X100. The mast cells were counted in five high-power fields (HPF) at X200.

### 2.12. Statistical Analysis

Values are expressed as means ± standard errors of means and were analyzed by one-way analysis of variance followed by Tukey’s multiple comparison *t*-test. The analysis was performed using GraphPad Prism software v4.0 (GraphPad Software Inc., La Jolla, CA, USA). Statistical significance was accepted for *p* values < 0.05. 

## 3. Results

### 3.1. Anti-Inflammatory Effects of Dictamine, Fraxinellone, Haplopine, and Obacunone on IL-6 mRNA Expression in HaCaT Cells

In order to investigate the effects of dictamine, fraxinellone, haplopine, and obacunone on cytokine release, we examined to determine whether they suppress IL-6 expression, a critical mediator of inflammation. IL-6 mRNA expression increased 6.6-fold after co-treating HaCaT-keratinocytes with IFN-γ/TNF-α. All four compounds reduced IL-6 mRNA expression in TNF-α/IFN-γ-stimulated HaCaT cells, but haplopine (25 μM) had the greatest inhibitory effect and inhibited IL-6 mRNA expression by 42%, which was more than that achieved by tacrolimus (38% inhibition) (Figure 2).

### 3.2. Inhibitory Effects of Haplopine on TSLP, GM-CSF, G-CSF, and IL-6 Expressions in HaCaT Cells

Since haplopine inhibited IL-6 mRNA expression most effectively, we investigated whether it suppresses the productions of the pro-inflammatory cytokines TSLP, G-CSF, and GM-CSF. TNF-α/IFN-γ co-treatment increased TSLP, GM-CSF, and G-CSF mRNA expressions by 3.5, 3.5, and 3.2-fold, respectively, versus vehicle controls. However, 12.5 μM haplopine pretreatment suppressed these TSLP, GM-CSF, and G-CSF increases to 2.7, 1.9, and 1.9-fold, respectively, versus vehicle controls. Furthermore, haplopine at 25 μM significantly reduced TSLP, GM-CSF, G-CSF increases to 1.8, 1.2, and 1.2-fold, respectively, versus vehicle controls, indicating haplopine inhibited the mRNA expressions of these three pro-inflammatory cytokines in a concentration-dependent manner (Figure 3A–C). The haplopine-associated attenuation of IL-6 and GM-CSF gene expressions were accompanied by a decrease in the level of the corresponding proteins (Figure 3D,E).

### 3.3. Inhibitory Effects of Haplopine on the Expressions of IL-4, IL-13, and COX-2 in Jurkat T Cells

Treatment with TNF-α/IFN-γ strongly increased the expressions of IL-4, IL-13, and COX-2 mRNAs in Jurkat T cells. Haplopine pretreatment at 25 and 50 µM attenuated TNF-α/IFN-γ-induced upregulation of IL-4 (Figure 4A). Haplopine also inhibited TNF-α/IFN-γ-induced IL-13 expression by up to 63% at 25 μΜ and 89% at 50 μΜ, versus that observed in TNF-α/IFN-γ-treated cells to levels lower than that observed in the control group (Figure 4B). Haplopine inhibited the TNF-α/IFN-γ-induced upregulation of COX-2 mRNA expression (Figure 4C) by 70% at 25 μM haplopine and by 100% inhibition at 50 μM versus that observed in TNF-α/IFN-γ-treated cells. Haplopine also inhibited the expression of COX-2 at the protein level in a concentration-dependent manner (Figure 4D).

### 3.4. Inhibitory Effects of Haplopine on SOD, CAT, and HO-1 Activities in Jurkat T Cells

Figure 5 shows the antioxidative effect of haplopine on the activation of SOD and CAT, which are the first line enzymes of the antioxidant defense system. The treatment with H_2_O_2_ significantly reduced the activation of SOD and CAT to 0.68 and 0.53-fold, respectively, versus vehicle controls. Cotreatment with H_2_O_2_ and haplopine increased SOD activity to 0.86-fold at 25 μM and to 0.91-fold at 50 μM (Figure 5A). Our results also showed a significant increase in the CAT activity in the haplopine groups. CAT activity was increased to 0.84-fold of control by treatment with 50 μM haplopine (Figure 5B). The HO-1 of the NC group was reduced to 0.48-fold compared to the CON group after H_2_O_2_ treatment. Haplopine (25 μM and 50 μM) treatment increased HO-1 levels to 0.79 and 0.88-fold, respectively, versus vehicle controls (Figure 5C). In addition, the decreased protein expressions of SOD, CAT, and HO-1 in H_2_O_2_-stimulated Jurkat T cells were restored by haplopine (12.5, 25, and 50 μM) treatment (Figure 5D).

### 3.5. Effect of Haplopine on AD-Like Skin Lesions in the DNCB-Induced Animal Model

On the last day of the experiment, severe AD-like skin symptoms including erythema, hemorrhage, edema, excoriation, erosion, dryness, and scarring were observed in the NC group (Figure 6A). The ameliorative effect of haplopine on the development of these AD-like symptoms was confirmed by lower clinical dermatitis scores (Figure 6B). Three weeks after DNCB sensitization, the mean clinical dermatitis score was significantly higher in the NC group than in the CON group. When haplopine was administered at concentrations of 0.05% and 0.1%, clinical dermatitis scores were significantly reduced by 50% and 40%, respectively, of those observed in the NC group.

### 3.6. Effects of Haplopine on Total Serum IgE and Splenic Cellularity

As shown in Figure 7A, repeated topical application of DNCB increased serum IgE concentration by >20-fold. Dermal application (0.05% or 0.1%) of haplopine markedly decreased the total serum IgE levels by 23% and 41%, respectively, as compared with the NC group. Furthermore, DNCB treatment increased spleen weights. Positive controls showed no significant change over the experimental period, but the 0.05% and 0.1% haplopine groups exhibited dose-dependent decreases in spleen weight as compared with the NC group (Figure 7B).

### 3.7. Effects of Haplopine on Histologic Manifestations in DNCB Treat Mice

To investigate the effect of haplopine on DNCB-induced mast cell infiltration and epidermis thickness, tissue sections were stained with toluidine blue or H&E. The number of mast cells stained with toluidine blue was significantly greater (>20-fold) in the NC group than in the CON group. However, treatment with 0.05% or 0.1% haplopine reduced the number of infiltrating mast cells by 61% and 52%, respectively (Figure 8A,C). Epidermal thicknesses were significantly greater in the NC group than in the CON group (by >3-fold). Treatment with 0.05% or 0.1% haplopine reduced epidermal thickening by 50% and 60%, respectively, as compared with the CON group (Figure 8B,D).

## 4. Discussion

AD is a representative chronic inflammatory skin disease with a complex etiology [22] and is characterized by skin barrier defects, immune dysregulation, and an increased risk of skin infections [23,24]. Recently, oxidative stress was suggested to be involved in the pathogenesis of AD [25], which triggers cutaneous inflammation by inducing epidermal keratinocytes to release pro-inflammatory cytokines and compromises skin barrier function [25,26]. For this reason, antioxidants are considered beneficial for the prevention and/or treatment of AD. 

*Dictamnus dasycarpus* is widely distributed in Asia and has been shown to possess anti-atopic [14], antioxidant [15], anti-inflammatory [15], and anti-allergic [14] properties. Furthermore, *D. dasycarpus* contains several bioactive alkaloids, limonoids, and terpenoids [19,20,27], and its extract has been reported to alleviate oxazolone-induced atopic-like dermatitis in mice [14]. The present study was performed to identify biologically active components in *D. dasycarpus* with anti-inflammatory, antioxidant, and/or anti-AD effects by performing in vitro and in vivo experiments in TNF-α/IFN-γ-induced HaCaT cells, H_2_O_2_-treated Jurkat T cells, and in a murine DNCB-induced model of AD. Fraxinellone, dictamine, and obacunone have been shown to be produced by *D. dasycarpus* [19,20,27], and haplopine isolated from this plant has been reported to have strong anti-inflammatory and anti-fungal properties [18,20,21]. Therefore, we selected these four compounds and investigated their potentials with respect to F. 

We confirmed all four compounds inhibited IL-6 mRNA expression and found that haplopine (25 µM) most actively suppressed TNF-α/IFN-γ-induced IL-6 expression in HaCat cells to the same extent as tacrolimus (the positive control) at the same concentration. Haplopine was also markedly suppressed the expressions of GM-CSF, G-CSF, and TSLP in a concentration-dependent manner in TNF-α/IFN-γ stimulated HaCat cells. Epidermal keratinocytes are the main cellular constituents of the epidermis and may contribute to the pathogenesis of AD by producing pro-inflammatory genes [28], and a number of studies have demonstrated that keratinocytes produce TNF-α, IFN-γ, and IL-6, which are considered to be crucial mediators of inflammation [29,30]. Furthermore, the keratinocytes of AD skin lesions overproduce GM-CSF, G-CSF, and TSLP, and TSLP plays an important role in the inflammation associated with atopic diseases [31,32]. Our results showed haplopine effectively inhibited the expressions of IL-6, GM-CSF, G-CSF, and TSLP in human keratinocytes, which suggests it has therapeutic potential as an anti-AD agent.

Our in vivo study was performed using DNCB topically treated Balb/c mice with AD-like skin lesions [33]. Histopathological analysis confirmed that haplopine treatment reduced mast cell infiltration and DNCB-induced epidermal thickening and thus, alleviated atopic skin symptoms in DNCB-induced mice. In addition, haplopine at 0.05% or 0.1% significantly reduced DNCB-induced serum IgE level increases to the same extent as 0.1% tacrolimus-treated positive controls. AD skin is characterized by the overexpression of IgE, and IgE-mediated mast cell and eosinophil activations are known to contribute to the pathogenesis and progression of AD [34,35], and thus, IgE is considered as a major target for alleviating atopic symptoms [36]. Previous clinical studies have also shown anti-IgE therapy has several therapeutic effects in AD [37]. According to our findings, haplopine markedly down-regulates the serum levels of IgE, has strong anti-inflammatory effects, and significantly inhibits the development of AD-like skin lesions.

Haplopine inhibited the expressions of IL-4, IL-13, and COX-2, which were increased by oxidative stress in H_2_O_2_-treated Jurkat T cells. Dysregulation of antioxidant mechanisms contributes to oxidative stress, which is caused by an imbalance between ROS generation and activity of the antioxidant defense system [38,39,40], and reportedly, excessive oxidative stress is a major etiological factor of AD because it induces inflammatory genes such as IL-4, IL-13, and COX-2 [41,42,43,44]. Therefore, inhibition of oxidative stress may importantly alleviate AD by downregulating inflammatory mediator levels. We found haplopine effectively inhibited the gene expressions of IL-4, IL-13, and COX-2 and significantly induced the activations of SOD, CAT, and HO-1 in H_2_O_2_-treated Jurkat T cells. SOD, CAT, and HO-1 are representative antioxidant enzymes considered to be principal negative mediators of oxidative stress [45,46,47]. Thus, because oxidative stress is considered an important contributor to the onset of AD, activation of the antioxidant system is considered an important anti-AD strategy [11]. Accordingly, our findings suggest haplopine may provide a means of treating AD as an antioxidant since antioxidants are prominent candidates for AD prevention and/or treatment. 

## 5. Conclusions

Haplopine, one of the biologically active compounds found in *D. dasycarpus*, effectively inhibited TNF-α/IFN-γ-induced upregulations of IL-6, TSLP, GM-CSF, and G-CSF in HaCat cells, and markedly reduced the H_2_O_2_-induced upregulations of IL-4, IL-13, and COX-2 and increased the activities of the antioxidant enzymes SOD, CAT, and HO-1 in Jurkat T cells. Furthermore, topical application of haplopine alleviated DNCB-induced AD-like symptoms in Balb/c mice and decreased mast cell infiltration and serum IgE concentrations in lesioned skin. These results suggest that haplopine should be considered a potential potent, natural treatment for AD with anti-inflammatory and antioxidant effects.

## Figures and Tables

**Figure 1 antioxidants-10-00806-f001:**
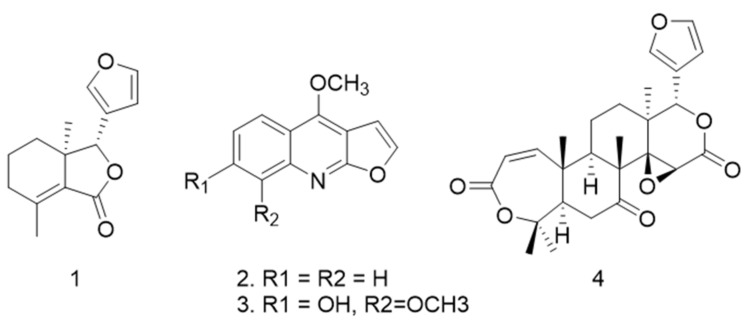
The four compounds of *Dictamnus dasycarpus* examined: (**1**) fraxinellone, (**2**) dictamine, (**3**) haplopine, and (**4**) obacunone.

**Figure 2 antioxidants-10-00806-f002:**
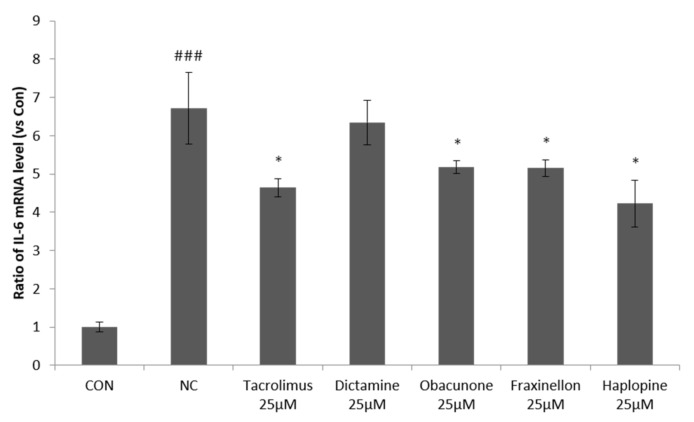
The effects of fraxinellon, dictamine, haplopine, and obacunone on IL-6 mRNA expression in TNF-α/IFN-γ-stimulated HaCaT cells. The effects of the four compounds on IL-6 mRNA expression were determined by quantitative Real-Time PCR relative to GAPDH. Values are expressed as the means ± SDs (n = 3). ^###^ *p* < 0.001 versus vehicle controls. * *p* < 0.05 versus TNF-α/IFN-γ-treated controls. CON: vehicle control; NC: TNF-α/IFN-γ-treated negative control; Tacrolimus 25 μM: TNF-α/IFN-γ + tacrolimus 25 μM-treated cells; Dictamine 25 μM: TNF-α/IFN-γ + dictamine 25 μM-treated cells; Obcunone 25 μM: TNF-α/IFN-γ + obacunone 25 μM-treated cells; Fraxinellon 25 μM: TNF-α/IFN-γ + fraxinellon 25 μM-treated cells; Haplopine 25 μM: TNF-α/IFN-γ + haplopine 25 μM-treated cells.

**Figure 3 antioxidants-10-00806-f003:**
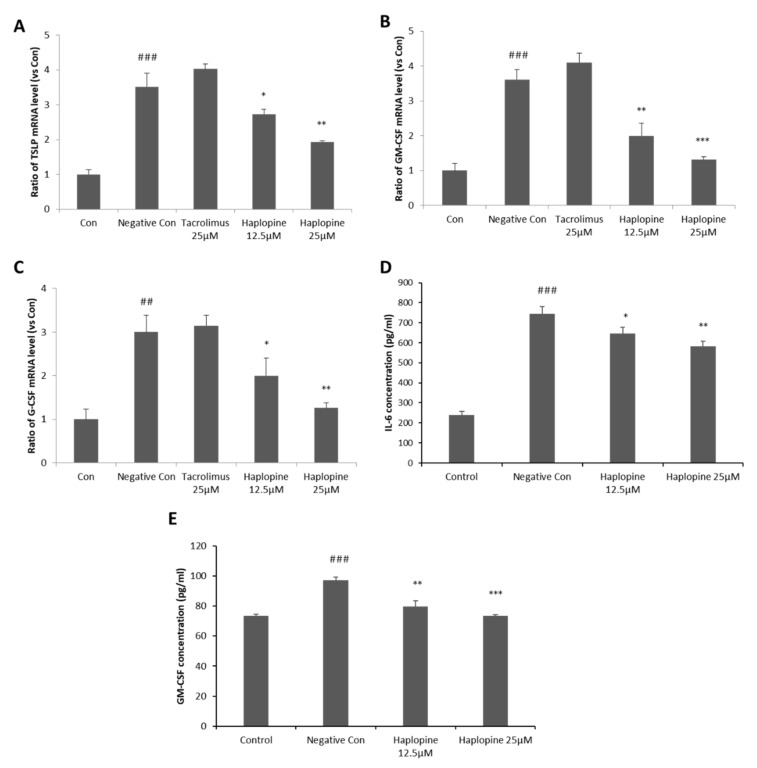
Inhibitory effects of haplopine on the mRNA expressions of TSLP, GM-CSF, and G-CSF mRNA and the protein expressions of IL-6 and GM-CSF in TNF-α/IFN-γ-stimulated HaCaT cells. (**A**) TSLP mRNA expression, (**B**) GM-CSF mRNA expression, and (**C**) G-CSF mRNA expression were analyzed by quantitative Real-Time PCR, and (**D**) IL-6 protein expression and (**E**) GM-CSF protein expression were analyzed using a spike glycoprotein enzyme-linked immunosorbent assay (ELISA). Values are expressed as the means ± SDs of three determinations. ^##^ *p* < 0.01 and ^###^ *p* < 0.001 versus vehicle controls. * *p* < 0.05, ** *p* < 0.01 and *** *p* < 0.001 versus TNF-α/IFN-γ-treated controls. CON: vehicle control; NC: TNF-α/IFN-γ-treated negative control; Tacrolimus 25 μM: TNF-α/IFN-γ + tacrolimus 25 μM-treated cells; Haplopine 12.5 and 25 μM: TNF-α/IFN-γ + haplopine 12.5 and 25 μM-treated cells.

**Figure 4 antioxidants-10-00806-f004:**
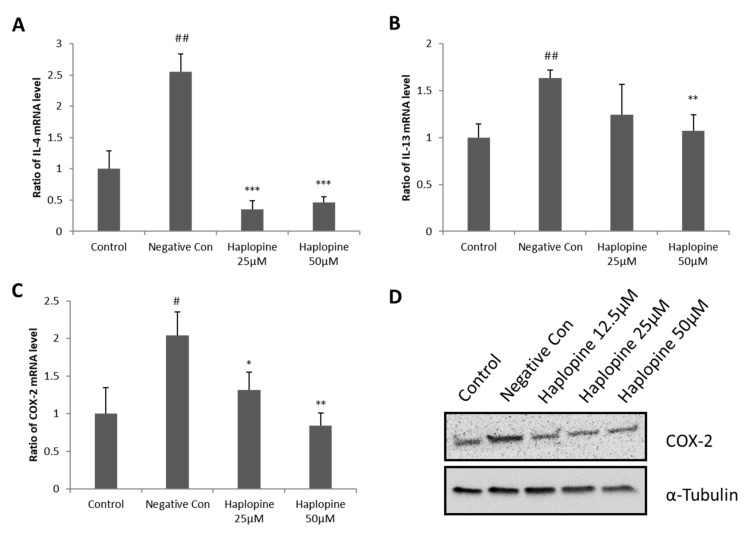
Inhibitory effects of haplopine on H_2_O_2_-induced IL-4, IL-13, and COX-2 mRNA expressions and COX-2 protein expression in Jurkat T cells. (**A**) IL-4, (**B**) IL-13, and (**C**) COX-2 mRNA expressions were analyzed by quantitative real-time PCR, and (**D**) COX-2 protein expression was analyzed by Western blot analysis. Values are expressed as the means ± SDs of three determinations. ^#^ *p* < 0.05 and ^##^ *p* < 0.01 versus vehicle controls. * *p* < 0.05, ** *p* < 0.01 and *** *p* < 0.001 versus H_2_O_2_-treated controls. CON: vehicle controls; NC: H_2_O_2_-treated negative control; Haplopine 12.5, 25, and 50 μM: H_2_O_2_ + haplopine at 12.5, 25, and 50 μM.

**Figure 5 antioxidants-10-00806-f005:**
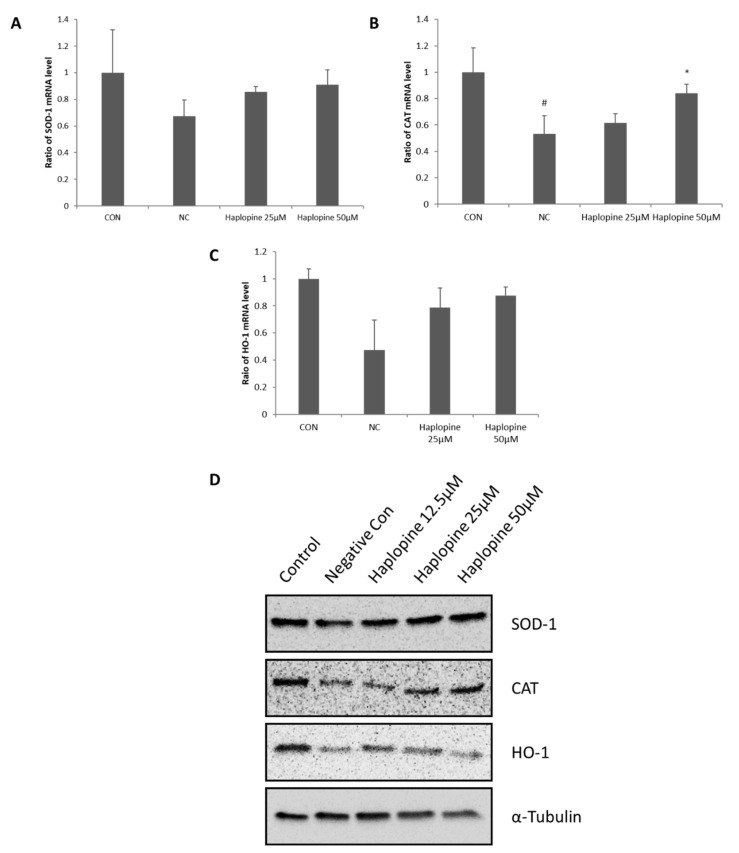
Effects of haplopine on the activations of antioxidant defense enzymes in H_2_O_2_-stimulated Jurkat T cells. (**A**) SOD, (**B**) CAT, and (**C**) HO-1 mRNA expressions were analyzed by real-time PCR, and (**D**) SOD, CAT, and HO-1 protein expressions were analyzed by Western blot analysis. Values are expressed as the means ± SDs of three determinations. ^##^ *p* < 0.01 and ^###^ *p* < 0.001 versus vehicle controls. * *p* < 0. relative mRNA values were normalized versus GAPDH. Values are expressed as the means ± SDs of three determinations. ^#^ *p* < 0.05 vs. normal controls. * *p* < 0.05 vs. H_2_O_2_-treated controls. CON: vehicle control; NC: H_2_O_2_-treated negative control; Haplopine 12.5, 25, and 50 μM: H_2_O_2_ + haplopine 12.5, 25, and 50 μM-treated cells.

**Figure 6 antioxidants-10-00806-f006:**
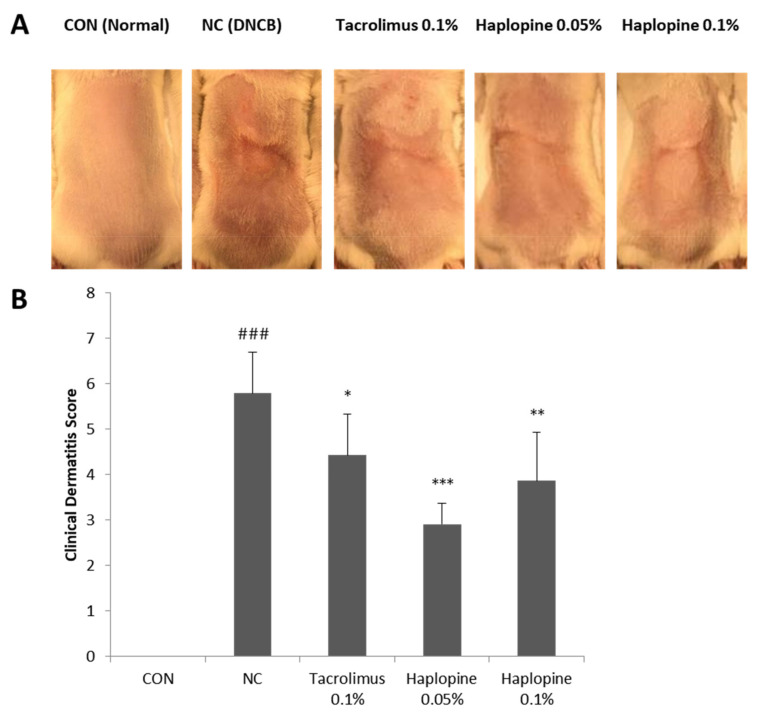
The effects of haplopine on clinical dermatitis scores in DNCB-induced BALB/c mice. Clinical dermatitis evaluations were performed in BALB/c mice after DNCB sensitization for 3 weeks. Photographs of (**A**) AD-like lesions in dorsal skin and (**B**) clinical dermatitis scores of AD-like skin lesions. Values are expressed as means ± SDs (n = 7). ^###^ *p* < 0.001 vs. the CON group.* *p* < 0.05, ** *p* < 0.01 and *** *p* < 0.001 vs. the NC group. CON: vehicle controls; NC: DNCB-treated controls; Tacrolimus 0.1%: DNCB + tacrolimus 0.1%-treated mice; Haplopine 0.05% and 0.1%: DNCB + haplopine 0.05% and 0.1%-treated mice.

**Figure 7 antioxidants-10-00806-f007:**
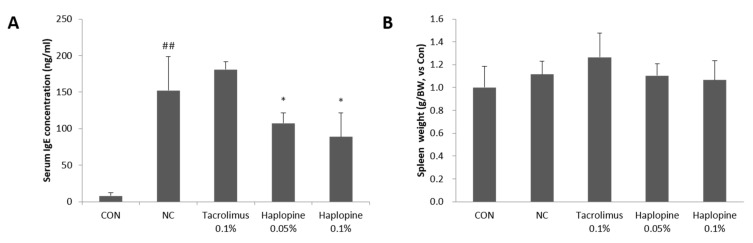
The effects of haplopine on the total serum IgE levels and spleen weights in DNCB-induced BALB/c mice. (**A**) The serum IgE levels were measured using an ELISA kit for IgE, and (**B**) spleen weights were measured using an electric balance. Values are expressed as means ± SDs (n = 7). ^##^ *p* < 0.01 vs. the CON group. * *p* < 0.05 vs. the NC group. CON: vehicle control; NC: DNCB-treated control; Tacrolimus 0.1%: DNCB + tacrolimus 0.1%-treated mice; Haplopine 0.05% and 0.1%: DNCB + haplopine 0.05% and 0.1%-treated mice.

**Figure 8 antioxidants-10-00806-f008:**
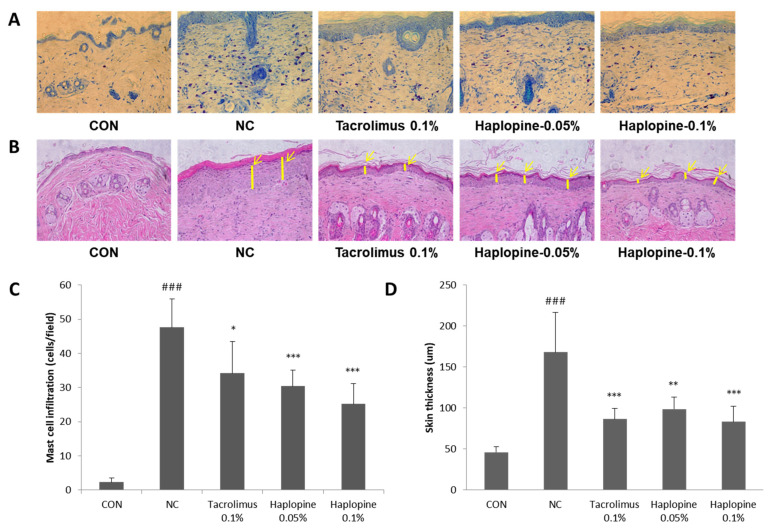
The effects of haplopine on histopathologic findings. Skin sections were stained with (**A**) toluidine blue or (**B**) H&E to analyze (**C**) mast cell infiltrations and (**D**) epidermal thicknesses. Values are expressed as means ± SDs (n = 7). ^###^ *p* < 0.001 vs. the CON group. * *p* < 0.05, ** *p* < 0.01 and *** *p* < 0.001 vs. the NC group. CON: vehicle control; NC: DNCB-treated control; Tacrolimus 0.1%: DNCB + tacrolimus 0.1%-treated mice; Haplopine 0.05% and 0.1%: DNCB + haplopine 0.05% and 0.1%-treated mice.

## Data Availability

The data presented in this study are available on request from the corresponding author.

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
