# Peer review of "Haplopine Ameliorates 2,4-Dinitrochlorobenzene-Induced Atopic Dermatitis-Like Skin Lesions in Mice and TNF-α/IFN-γ-Induced Inflammation in Human Keratinocyte"

_antioxidants, 2021, doi:10.3390/antiox10050806_

Round 1

Reviewer 1 Report

Authors of this paper aim to investigate the potential of haplopine, one of the active compounds found in D. dasycarpus, as a therapeutic agent in atopic dermatitis by analyzing its ability to act as an an anti-inflammatory and antioxidant mediator in TNF-α/IFN-γ induced HaCaT cells, H2O2 treated Jurkat T cells, and in a murine model of atopic dermatitis.

This study expands the knowledge of the anti-atopic dermatitis capacity of biologically active components in D. dasycarpus e to supress T cell immunity in autoimmune diseases by showing experimentally that haplotine is able to reduce serum levels of IgE and pruritus associated with eczematous atopic lesions in the murine model.

I hope the following comments will help them to improve the manuscript.

  1. General comment on the “Abstract”: The wording of the Abstract is confusing. It should be restructured to first present the general objective of the paper, which I assume is to study the anti-atopic dermatitis effects of haplopine. Then, it should be specified the three experimental approaches that have been carried out to analyze in vitro its effects on keratinocytes and T cells and to study its in vivo action in DNCB-stimulated Balb/c mice. In the Discussion section, the text between lines 353 and 356 perfectly reflects the general structure of the paper that should have been included in the abstract.
  2. Comment on the “Title”: The authors should rethink the title of the paper as it only reflects one of the contributions of the study. It should not only refer to the results in the murine model but should be more general in some way to avoid excluding the results in human cell lines.
  3. Lines 55-60: The sentence contained in these lines is too lengthy and unclear. It would be recommendable to modify its wording and divide it into two parts.
  4. Line 148. Please, correct “…Jurket T cells…”
  5. Figure 6: Figure 6B does not show the symbols describing whether there are significant differences in clinical dermatitis scores.

Reviewer 2 Report

The manuscript entitled “Effect of Haplopine on 2,4-Dinitrochlorobenzene-Induced 2 Atopic Dermatitis-Like Skin Lesions in Mice” submitted to Antioxidants by Dr. Kim and co-workers presents interesting findings on the effects of the bioactive compounds of Dictamnus dasycarpus Turcz such as dictamine, haplopine, fraxinellone and obacunone on cytokine secretion and antioxidant system in HaCat and Jukat T cell cultures as well as on atopic dermatitis inflammation induced by 2,4-dinitrochlorobenzene. Although the study has interest, the manuscript needs a profound review.

Introduction or discussion should report the relationship between oxidative stress and cytokine/arachidonic acid cascade pathways (doi: 10.1080/10715760000300301) as well as the relationship between cytokines and cyclooxygenase pathway (doi: 10.1016/s0006-2952(98)00056-2).

Line 101. Tacrolimus is used as positive control. This molecule should be presented more suitability.

Line 117-124. Please checking this paragraph. Several sentences appear two times.

Line 160. “primary antibodies for 24 h, washed with Tris-buffered saline with Tween-20, and incubated again with anti-mouse immunoglobulin G horseradish peroxidase-conjugated secondary antibodies” Antibody source, antibody dilution, antibody incubation condition should be described.

Line 187. “Scoring was based on the severities of erythema/hemorrhage, edema, excoria- tion/erosion, and dryness/scarring/inflammation of dorsal skin, which were scored as 0 or 1. Dermatitis scores were calculated by summing scores for these four signs (none = 0; mild = 1; moderate = 2; severe = 3)” However, Figure 6 shows means around 6. Considering material and methods and results text I cannot understand the dermatitis score method used in the study.

Line 206. “…inflammatory cells or mast cells,….” I believe that mast cells are inflammatory cells.

Line 218. “In order to investigate the anti-inflammatory effects of dictamine, fraxinellone, haplopine, and obacunone, we examined to determine whether they suppress IL-6 expression…” In this in vitro studies, anti-inflammatory effects were not assayed. Here, authors studied the effects of bioactive molecules on cytokine release.

Line 228. “ 3 determinations” What means? One assay performed in triplicate? N = 3 is too much poor in an in vitro assay. In vitro studies with N = 3 whereas in vivo studies were performed using N = 7. Why?

Figure 3. Control and negative control are confused. Please changed to clarify.

Figure 3A-C present tacrolimus findings but Figure 3D-F did not contain tacrolimus results. Why?

Figure 5. Control and negative control are confused. Please changed to clarify. In my opinion, NC group as Control and Control as H2O2 is clearer. The order of bars should be also corrected.

Figure 6B. Statistical significance symbols are not presented.

Line 360. “…to the treatment of inflammatory skin diseases.” This expression is too much ambiguous.

Line 375. “….haplopine treatment reduced mast cell infiltration and DNCB-induced epidermal thickening, and thus, alleviated….” What was epidermal thickening measured?

Line 384. “….and significantly inhibits pruritus associated with eczematous atopic lesions.” What was pruritus measured? These aspects of the discussion did not supported by the appropriate findings.

Line 398.  “Accordingly, our findings suggest haplopine may provide a means of treating AD as a strong antioxdant.” Reactive oxygen species concentrations should be measured to improve this conclusion. The study only demonstrated that haplopine modulates some antioxidant enzyme activities.

Line 406. “….decreased mast cell infiltration and serum IgE concentrations in lesioned skin.” What was IgE measured in lesioned skin? These findings were not presented in the manuscript.

The expression “dose-dependently” is used along but, in my opinion is not suitable. Concentration-dependent is more appropriate in in vitro assays. Furthermore, the study needs to assay 3-4 or more concentrations/doses to consider a dose/concentration-dependent effect. This should be corrected along the manuscript.

Checking typos such as antioxdant

Round 2

Reviewer 2 Report

The authors have considered my suggestions to improve the manuscript

This manuscript is a resubmission of an earlier submission. The following is a list of the peer review reports and author responses from that submission.